# A Systematic Review of the Adherence to Home-Practice Meditation Exercises in Patients with Chronic Pain

**DOI:** 10.3390/ijerph20054438

**Published:** 2023-03-02

**Authors:** Alberto Barceló-Soler, Héctor Morillo-Sarto, Selene Fernández-Martínez, Alicia Monreal-Bartolomé, Maria José Chambel, Paula Gardiner, Yolanda López-del-Hoyo, Javier García-Campayo, Adrián Pérez-Aranda

**Affiliations:** 1Institute of Health Research of Aragon (IIS), 50009 Zaragoza, Spain; 2Navarra Medical Research Institute (IdiSNA), 31008 Pamplona, Spain; 3Research Network on Chronicity, Primary Care and Health Promotion (RICAPPS), 50009 Zaragoza, Spain; 4Department of Psychology and Sociology, University of Zaragoza, 50009 Zaragoza, Spain; 5Department of Psychiatry, University of Zaragoza, 50009 Zaragoza, Spain; 6CicPsi, Faculdade de Psicologia, Universidade de Lisboa, 1649-004 Lisbon, Portugal; 7Center for Mindfulness and Compassion, Cambridge Health Alliance, Cambridge, MA 02141, USA; 8Department of Family Medicine, Medical School, University of Massachusetts, Worcester, MA 01655, USA; 9Department of Basic, Developmental and Educational Psychology, 08193 Cerdanyola del Vallès, Spain

**Keywords:** chronic pain, mindfulness, meditation, third wave psychotherapy, adherence, home practice, systematic review

## Abstract

Mindfulness-, compassion-, and acceptance-based (i.e., “third wave”) psychotherapies are effective for treating chronic pain conditions. Many of these programs require that patients engage in the systematic home practice of meditation experiences so they can develop meditation skills. This systematic review aimed at evaluating the frequency, duration, and effects of home practice in patients with chronic pain undergoing a “third wave” psychotherapy. A comprehensive database search for quantitative studies was conducted in PubMed, Embase, and Web of Sciences Core Collection; 31 studies fulfilled the inclusion criteria. The reviewed studies tended to indicate a pattern of moderately frequent practice (around four days/week), with very high variability in terms of time invested; most studies observed significant associations between the amount of practice and positive health outcomes. Mindfulness-Based Stress Reduction and Mindfulness-Based Cognitive Therapy were the most common interventions and presented low levels of adherence to home practice (39.6% of the recommended time). Some studies were conducted on samples of adolescents, who practiced very few minutes, and a few tested eHealth interventions with heterogeneous adherence levels. In conclusion, some adaptations may be required so that patients with chronic pain can engage more easily and, thus, effectively in home meditation practices.

## 1. Introduction

The International Association for the Study of Pain (IASP) defines chronic pain as pain that lasts or recurs for longer than three months, regardless of if it is experienced in one or more than one anatomical regions [1]. In some cases, chronic pain is linked to other diseases as the underlying cause (i.e., secondary chronic pain, as in cancer, post-surgical, posttraumatic, and neuropathic pain, among others), while in other cases, it is a disease of its own (i.e., primarily chronic pain: fibromyalgia, migraines, irritable bowel syndrome, and low back pain, among others) [2,3]. Chronic pain conditions very often have a significant negative impact on the individual’s functionality, and a notable majority of these patients present with anxiety and depression; all in all, chronic pain is a leading source of human suffering and disability [4].

Different therapeutic approaches are used for treating chronic pain conditions: pharmacotherapies (e.g., analgesics, tricyclic antidepressants, and opioids, among others) have resulted in mixed results for safety and effectiveness for managing chronic pain [5,6,7,8,9]. For some chronic pain conditions, different evidence-based guidelines do not agree on first-line recommendations of only pharmacotherapy, and some even give a higher ranking to cognitive-behavioral therapy (CBT) and multi-component treatments (i.e., interventions that include different strategies, such as psychoeducation, relaxation, physical exercise, etc.) [10,11]. Classic CBT for chronic pain has focused on changing negative thoughts and maladaptive behaviors about pain to improve function and coping [12]; although it has proved to be effective, the impact of CBT on chronic pain conditions has generally been small [13].

In the last decades, different programs based on CBT have been developed, although in this case, they are focused on promoting mindfulness, i.e., nonjudgmental, present-focused attention, and other related constructs, such as compassion and acceptance. These interventions are not primarily focused on symptom relief but, instead, on the promotion of well-being. This is accomplished through the cultivation of psychological flexibility (i.e., a wide range of abilities, such as recognizing and adapting to situational demands, shifting mindsets or behavioral repertoires when these strategies compromise personal or social functioning, maintaining a balance between life domains, and being aware, open, and committed to behaviors that are congruent with deeply held values) [14,15]. Meditation practices play a central role in these programs since they train the patient to interact with their thoughts and feelings in a more detached manner.

Such interventions have been categorized under the term “third wave” CBT psychotherapies; Mindfulness-Based Stress Reduction (MBSR) [16], Mindfulness-Based Cognitive Therapy (MBCT) [17], and Acceptance and Commitment Therapy (ACT) [18] are some examples. Previous studies have concluded that these interventions’ effects on reducing pain interference are achieved through enhancing pain acceptance and psychological flexibility [19], as well as reducing fear of pain, pain-related anxiety, and fear-avoidance beliefs [20]. These changes play a significant role in improving functional impairment, anxiety, depression, and psychological distress in chronic pain conditions [21,22,23,24,25]. Some studies report higher effects of “third wave” psychotherapies than multicomponent interventions [26], although the superiority of these interventions compared to classic CBT is still unclear [27].

In “third wave” psychotherapies, mindfulness is cultivated through meditations that are first experienced by the patient in live group-based sessions and intended to be practiced at home alone between sessions and beyond. Structured programs, such as MBSR or MBCT, which consist of a weekly 2–2.5 h session during 8 consecutive weeks, ask participants to engage in “formal” practices (i.e., guided meditations, such as body scan, Awareness of Breath meditation, and mindful movement with suggestions to posture, attitude, and how attention is directed) 6 days a week at home, investing around 45 min per practice [16,17], in addition to the normally brief and unstructured “informal” practices (i.e., applying mindfulness to everyday activities such as mindful eating or showering). Home practices are reviewed and discussed in the next session with the instructor. These autonomous home practices are assumed to be critical to therapeutic change: the meta-analysis conducted by Parsons et al. [28] observed a significant association between the extent of formal mindfulness practice and positive intervention outcomes for a wide range of conditions, such as depression, anxiety, or insomnia. However, despite the importance of home practice, there is great heterogeneity in the instructions and resources provided to the participants and the methodology used to monitor home practice, as well as in the recommended duration of the practices, in the scientific studies [29].

The present systematic review focuses on chronic pain conditions. These chronic conditions require an ongoing need for self-management skills, such as mindfulness, to have a long-term impact on the physiology of the brain and overall health, which requires systematic and frequent home practice. Parsons et al. [28] observed that across 43 studies (N = 1427), the pooled estimate for participants’ home practice was 64% of the assigned amount, equating to about 30 min per day, six days per week [95% CI 60–69%]. The amount of home practice could be substantially different when considering patients with chronic pain for various reasons, such as physical discomfort (e.g., people with fibromyalgia may have difficulties with long durations of standing or sitting, which is often required in formal meditations) or cognitive impairment that is often associated with some chronic pain syndromes [30,31]. These could represent potential barriers to engaging in home practices, which might hinder the intervention’s effectiveness. Thus, the present work will systematically review quantitative studies conducted on samples of patients with chronic pain conditions who underwent a “third wave” psychotherapy to determine the frequency and duration of home practice of mindfulness and/or other related skills. When reported, the impact of such practice on main symptoms will be discussed, and certain variables (e.g., type of chronic pain, type of intervention, age group) will be granted special attention.

## 2. Materials and Methods

### 2.1. Design

A systematic review of quantitative studies, including randomized controlled trials (RCTs), non-RCTs, open trials with pre-post analysis, and pilot studies, was conducted. This systematic review was registered in the PROSPERO database (Registration number: CRD42022351572) and was implemented in accordance with the standards of Preferred Reporting Items for Systematic Reviews and Meta-analysis (PRISMA) guidelines [32] (checklist can be found in Appendix A).

### 2.2. Data Sources and Search Strategy

A search strategy using PubMed, Embase, and Web of Sciences Core Collection databases was performed between September and October 2022. Only studies published in English were included, and no restrictions regarding publication dates were considered. Specific search terms (e.g., MeSH terms) were combined and adapted to each database and are detailed in the Appendix A.

Also, relevant studies were sought from reference lists of included studies. The titles and abstracts of all potentially eligible studies identified from the search were reviewed against the inclusion and exclusion criteria by three reviewers. The full texts of all potential studies were then independently screened by three reviewers, and disagreements were resolved through discussion with a fourth reviewer.

### 2.3. Elegibility Criteria

The following inclusion and exclusion criteria were guided by the evidence-based medicine PICOS framework or strategy [33]:Participants: Patients who suffer from chronic pain conditions, both primary (e.g., fibromyalgia, irritable bowel syndrome, migraines, etc.) and secondary chronic pain (e.g., post-surgical pain, rheumatoid arthritis, chronic pain associated with a medical condition, etc.);Interventions and comparisons: The interventions included in this review were those constituted under the epigraph of “third wave” CBT (based on the practice of mindfulness, compassion, acceptance, etc.) and which prescribe the continued practice of the exercises at home by the participants. Interventions could be individual or group-based and could be conducted either in person or online. Blended interventions (i.e., a combination of online and in-person sessions) were also included. The duration of the intervention could vary. There were no inclusion/exclusion criteria regarding comparators/controls;Outcomes: Home practice frequency (i.e., number of days) and amount of practice (i.e., minutes per day) were the main outcomes of the systematic review. When reported, the relationship between practice and the study variables (e.g., pain, functionality, depression, anxiety, etc.) was also presented as an outcome. Information on home practice time had to cover the duration of the course and/or a post-intervention follow-up time. When detailed, practice time was reported for both so-called formal and informal practices. Barriers to or facilitators of maintaining the habit of practicing at home were included as secondary outcomes.Study design: We included randomized controlled trials (RCTs), non-RCTs, and open trials with pre-post analysis. Pilot studies were also included as long as they followed one of the previous study designs. Studies published in peer-reviewed journals as well as pre-print papers, were included. On the other hand, cross-sectional studies, qualitative studies, study protocols, reviews, clinical cases, conference proceedings, letters, commentaries, and case studies were excluded.

### 2.4. Search Outcomes

Initially, 9102 articles were retrieved from the databases PubMed, Embase, and Web of Sciences Core Collection; in addition, one article was identified from the citation search. Once duplicates were removed, a total of 5297 articles were identified and screened by three reviewers (A.B.-S., S.F.-M., and A.P.-A.); the full-text manuscripts for 157 studies were then sought for retrieval and assessed, of which 31 met the inclusion criteria (Figure 1).

### 2.5. Quality Appraisal

To assess the quality of the studies, we used the assessment tool developed by National Heart, Lung, and Blood Institute, one for controlled intervention studies and one for single-arm studies. This allowed the included studies, both controlled intervention studies, and single-arm studies, to be evaluated for risk of bias.

Two reviewers (A.M.-B. and A.B.-S.) conducted such a quality assessment of the included studies, and disagreements were resolved through discussion between the researchers and in consultation with a third author (A.P.-A.). The quality assessment is provided in the Appendix A along with a link to the assessment tools and their items.

### 2.6. Data Abstraction

Selection and coding were performed independently by three authors (A.B.-S., S.F.-M., and A.P.-A.). All detected articles were screened according to the established inclusion criteria; then, abstracts and titles were reviewed to identify those relevant to the research question. When too little information was available to determine eligibility, full articles were reviewed. Relevant articles were subsequently selected by cross-examination of articles. Disagreements between authors were resolved by consensus, and when in doubt, the final decision was made in consultation with a fourth author (H.M.-S.). A data extraction form was developed based on Centre for Reviews and Dissemination templates. Data collected included information on authors, date of publication, country of study, study design, sample characteristics (e.g., sample size, chronic pain condition, age), intervention type (i.e., MBSR, MBCT, ACT, etc.) and its duration, intervention format (e.g., face-to-face, online, blended), home practice details (frequency in days and time in minutes), and primary and secondary outcomes measured.

## 3. Results

### 3.1. Summary of the Findings

Table 1 provides a summary of the 31 studies included in this systematic review. The years of publication of the studies ranged between 2007 and 2022. Of the included studies, eight (26%) were conducted in Europe (United Kingdom, Spain, Denmark, and The Netherlands), twenty (65%) in America (United States of America and Canada), two (6%) in Oceania (Australia), and one (3%) in Asia (Hong Kong). Regarding the study design, twenty-four (78%) were RCTs, two (6%) non-RCTs, four (13%) were single-arm studies, and one (3%) was a secondary analysis of a previous RCT. With respect to the type of demographics included in the studies, four (13%) were carried out with a population that was 18 years old or younger, and twenty-seven (87%) with an adult population. Additionally, in four (13%) studies, the sample was made up only of the female population; in twenty-three (74%), it was made up of both men and women, and in four (13%), this information was not reported.

The most prevalent types of chronic pain conditions included fibromyalgia, irritable bowel syndrome, chronic low back pain, migraine, and medical conditions with chronic pain as a symptom (i.e., secondary chronic pain) (Table 1). Among the interventions studied, fourteen studies used standardized MBSR or MBCT programs. The other interventions combined mindfulness with other elements, such as acceptance, CBT techniques, and yoga exercises, among others. Despite the heterogeneity between the interventions, twenty-seven (88%) studies tested interventions with a duration of eight weeks. The duration of the remaining interventions ranged between 3 and 12 weeks.

An aspect of interest in this type of study is the way in which home practices are assigned. On the one hand, studies that used the Internet as a resource to facilitate audio or video with guided practices stand out, such as through YouTube [34], email [35], web pages developed specifically for the study [36,37], and download platforms [38]. On the other hand, some studies used physical books, CDs, and DVDs, and in which the practices that the participants had to carry out at home were compiled [26,39,40,41,42,43,44,45,46,47]. However, despite the relevance of this aspect, some studies did not specify how the participants received the instructions and resources to practice between sessions [48,49,50,51,52,53,54,55,56,57,58].

The way in which the frequency of home practice was recorded varies between the included studies. Only one study used both daily (during the intervention) and weekly (during follow-up) recordings of the home practice [41]. Fourteen studies used daily logs, of which two studies specified doing so online [35,52]; another two studies used pen and paper [55,59]; one study used both depending on the resources of the participants [42]; the others did not specify how they collected information [26,34,43,44,49,53,56,60,61]. Three studies used the weekly logs as a strategy to collect practice time at home [39,40,48]. On the other hand, some studies used other strategies to collect home practice but without specifying the timing. One study collected the information through a qualitative interview once the intervention had finished [62]; two studies created an ad hoc item for home practice time that was included as part of a post-intervention questionnaire [37,54]; one study used a logbook created specifically for the study but without specifying its characteristics [57]; and another study obtained the information from the number of logs turned in [50]. Finally, eight studies did not specify any information regarding the methodology used to collect the information [36,38,45,46,47,51,58,63].

**Table 1 ijerph-20-04438-t001:** Description of the studies reviewed.

Authors (Year)	Design	Sample	Chronic Pain Condition	Intervention (Duration)
Ali et al. (2017) [62]	Single-arm study (3 cohorts)	N = 18 (11 females) Age: 14.80 (range: 10–18)	Functional somatic syndromes	MBSR (8 weeks)
Carson et al. (2010) [42]	RCT (2 arms: Yoga of Awareness Program vs. wait-list control)	N_Yoga-Awareness_ = 25 (all females) Age: 51.40 (SD = 13.17)	Fibromyalgia	Yoga of Awareness program (8 weeks)
Cebolla et al. (2021) [37]	Non-randomized controlled trial (2 arms: MBI vs. standard medical therapy)	N_MBI_ = 45 (43 females) Age: 45.41 (SD = 10.46)	Inflammatory Bowel Disease	Blended MBI (8 weeks)
Chadi et al. (2016) [57]	RCT (2 arms: MBI vs. wait-list control)	N_MBI_ = 19 (all females) Age: 15.80 (SD = 1.10)	Chronic pain as a result of a medical condition	Adapted MBI (8 weeks)
Chadi et al. (2018) [54]	RCT (2 arms: online MARS-A vs. in person MARS-A)	N = 18 (14 females) Age: 15.30 (range 13–18)	Medical condition that implies chronic pain or headaches	MARS-A Program (8 weeks)
Cooperman et al. (2021) [49]	RCT (2 arms: MORE vs. TAU)	N_MORE_ = 15 (8 females) Age: 47.90 (SD = 8.70)	Patients with opioid use disorder and primary chronic pain	MORE (8 weeks)
Day et al. (2014) [58]	RCT (2 arms: MBCT vs. TAU)	N_MBCT_ = 19 (17 females) Age: 43.10 (SD = 11.20)	Primary headache pain (and other comorbid chronic pain conditions)	MBCT (8 weeks)
Day et al. (2016) [52]	Secondary analysis of Day et al. (2014)	N = 21 (20 females) Age: 42.80 (SD = 12.50)	Primary headache pain (and other comorbid chronic pain conditions)	MBCT (8 weeks)
Day et al. (2020) [55]	RCT (3 arms: CT vs. MM vs. MBCT)	N_MM+MBCT_ = 56 (sex n.r.) Age: 50.74 (SD = 14.43)	Chronic low back pain	MBCT (8 weeks)
Donnino et al. (2021) [48]	RCT (3 arms: MBSR vs. PSRT vs. TAU)	N_MBSR_ = 12 (6 females) Age: 39.30 (SD = 14.40)	Chronic back pain	MBSR (8 weeks)
Gardiner et al. (2020) [36]	Single arm study (2 cohorts)	N = 43 (39 females) Age: 50.40 (SD = 12.60)	Chronic pain (as a symptom)	Our Whole Lives for Chronic Pain (9 weeks)
Garland et al. (2014) [45]	RCT (2 arms: MORE vs. support group)	N_MORE_ = 57 (sex n.r.) Age: 48 (SD = 14)	Patients with opioid use disorder and chronic pain	MORE (8 weeks)
Greenberg et al. (2019) [50]	Non-randomized controlled trial (2 arms: 3RP + GetActive vs. 3RP + GetActive with Fitbit)	N = 13 (10 females) Age: 44 (SD = 14.31)	Different chronic pain conditions	3RP (8 weeks)
Hearn and Finlay (2018) [63]	RCT (2 arms: online MBI vs. online psychoeducation)	N_MBI_ = 36 (19 females) Age: 43.80 (SD = 8.70)	Patients with chronic pain after spinal cord injury	Online MBI (8 weeks)
Hesse et al. (2015) [59]	Single arm study	N = 20 (all females) Age: 14.15 (SD = 1.60)	Recurrent headaches	Adapted Mindful School Curriculum for Adolescents (8 weeks)
Howarth et al. (2019) [60]	RCT (2 arms: Brief MBI vs. active control)	N_MBI_ = 37 (24 females) Age: 54.70 (SD = 12.50)	Persistent pain (as a symptom)	Brief MBI (4 weeks)
Johannsen et al. (2018) [51]	RCT (2 arms: MBCT vs. wait-list control)	N_MBCT_ = 67 (all females) Age: 56.80 (SD = 9.99)	Persistent pain (as a symptom of breast cancer)	MBCT (8 weeks)
Mittal et al. (2022) [35]	RCT (2 arms: MBCT vs. TAU)	N_MBCT_ = 22 (15 females) Age: 54.20 (SD = 12.80)	Persistent chest pain (non-cardiac cause)	MBCT (8 weeks)
Morone et al. (2008) [43]	RCT (2 arms: MBSR vs. wait-list control)	N_MBSR_ = 19 (10 females) Age: 74.10 (SD = 6.10)	Chronic low-back pain	MBSR (8 weeks)
Pérez-Aranda et al. (2019) [26]	RCT (3 arms: MBSR vs. FibroQoL vs. TAU)	N_MBSR_ = 75 (73 females) Age: 52.96 (SD = 7.98)	Fibromyalgia	MBSR (8 weeks)
Pradhan et al. (2007) [61]	RCT (2 arms: MBSR vs. TAU)	N_MBSR_ = 31 (26 females) Age: 56 (SD = 9)	Rheumatoid arthritis	MBSR (8 weeks)
Rae et al. (2020) [46]	RCT (2 arms: MBI + yoga vs. stretching class)	N_MBI_ = 10 (2 females) Age: 51.70 (SD = 14.90)	Chronic low-back pain	MBI + yoga (8 sessions)
Rosenzweig et al. (2010) [44]	Single-arm study	N = 133 (111 females) Age: 52.96 (SD = 7.98)	Different chronic pain conditions (mostly primary chronic pain)	MBSR (8 weeks)
Seng et al. (2019) [56]	RCT (2 arms: MBCT vs. TAU/wait-list control)	N = 31 (29 females) Age: 36.20 (SD = 10.60)	Migraine	MBCT (8 weeks)
Trompetter et al. (2014) [38]	RCT (3 arms: online ACT vs. online expressing writing vs. wait-list control)	N_ACT_ = 82 (63 females) Age: 52.90 (SD = 13.30)	Chronic pain (as a symptom)	Online ACT (9–12 weeks)
Van Gordon et al. (2017) [47]	RCT (2 arms: MAT vs. CBT)	N_MAT_ = 74 (61 females) Age: 46.41 (SD = 9.06)	Fibromyalgia	MAT (8 weeks)
Wong et al. (2011) [39]	RCT (2 arms: MBSR vs. MPI)	N_MBSR_ = 51 (sex n.r.) Age: 48.70 (SD = 7.84)	Chronic pain (as a symptom)	MBSR (8 weeks)
Zanca et al. (2022) [34]	RCT (2 arms: CMI vs. active control)	N_CMI_ = 11 (2 females) Age: 50 (range 37–65)	Patients with chronic pain after spinal cord injury	CMI (4 weeks)
Zernicke et al. (2012) [40]	RCT (2 arms: MBSR vs. wait-list control)	N_MBSR_ = 43 (40 females) Age: 45 (SD = 12.40)	Irritable bowel syndrome	MBSR (8 weeks)
Zgierska et al. (2016) [41]	RCT (2 arms: MBI vs. TAU)	N_MBI_ = 21 (15 females) Age: 52.70 (SD = 10.50)	Chronic low back pain	MBI (8 weeks)
Zgierska et al. (2016) [53]	RCT (2 arms: MM + CBT vs. TAU)	N_MM+CBT_ = 21 (sex n.r.) Age: 51.80 (SD = 9.70)	Patients with opioid-treated chronic low back pain	MM + CBT (8 weeks)

Abbreviations: CMI = Clinical Meditation and Imagery; CT = Cognitive Therapy; MARS-A = Mindful Awareness and Resilience Skills for Adolescents; MAT = Meditation awareness training; MM = Mindfulness Meditation; MM + CBT = Mindfulness Meditation and Cognitive Behavioral Therapy; MORE = Mindfulness-Oriented Recovery Enhancement; MPI = Multidisciplinary Pain Intervention; PSRT = Psychophysiologic Symptom Relief Therapy; TAU = Treatment As Usual; 3RP = Mind–body Relaxation Response Resiliency Program; n.r. = “not reported”.

### 3.2. Study Quality

The overall quality of the 31 included studies was rated as fair. Only 10 (32%) included studies (10/27 controlled studies) were rated as good, while 12 (39%) (9/27 controlled studies and 3/4 single-arm studies) were rated as poor. For controlled studies, the risk of bias was mainly due to lack of blinding, the differences between groups at baseline, the high dropout rate at the endpoint (>20%), or the lack of sample size/power calculation. For the single-arm studies, the risk of bias was related to the lack of information on the sample’s representativeness, the non-inclusion of all the participants who met the inclusion/exclusion criteria, no sample size calculation, lack of blindness of people assessing outcomes, and the lack of follow-up measures.

### 3.3. Frequency of Home Practice

Table 2 displays the information on each reviewed study. Sixteen studies (51.6%) reported the number of days that patients with chronic pain practiced meditation at home during the intervention: on average, participants engaged in home practices for 4.27 days per week (SD = 1.40). In addition, twenty-four studies (77.4%) reported the minutes of practice: 26.78 min per day (SD = 21.10) during the intervention. Only four studies [48,53,56,60] reported the frequency of practice in follow-up assessments (1 to 4 months), finding an average frequency of 3.49 days per week (SD = 1.05) and 32.23 min per day (SD = 16.73).

### 3.4. Impact of Home Practice on Main Symptoms

Nine studies explored the impact of the amount of practice on health outcomes, and seven of them found significant effects: Ali et al. [62] and Carson et al. [42] found that frequency of practice was significantly associated with improvement in functionality, fatigue, and relaxation. Day et al. [58] found that MBCT completers improved significantly on pain catastrophizing, pain acceptance, and self-efficacy compared to TAU and, similarly, Zgierska et al. [41] observed that “consistent meditators” had a greater decrease in pain ratings when compared with controls. In the same vein, Johannsen et al. [51] found that more work practice during the intervention predicted increases in mindfulness non-reactivity but not in pain catastrophizing (and no effects in the follow-up were observed). Rosenzweig et al. [44] reported a significant impact of the amount of home meditation practices on psychological distress, somatization symptoms, and self-rated general health. Similarly, Van Gordon et al. [47] also found that the number of minutes of meditation was significantly related to functionality, pain, psychological distress, sleep quality, and non-attachment.

On the other hand, two studies did not find any significant impact of the frequency of practice on the study outcomes: Day et al. [55] observed medium, yet not statistically significant, effects on pain interference and pain intensity, and Pradhan et al. [61] found that neither the overall sum of practice time nor the sum of time spent on a specific practice predicted change in any measure by two months, although each one-day increase in practice was associated with improvement in depressive symptoms and in psychological distress.

## 4. Discussion

Home meditation practice is considered a key part of “third wave” CBT psychotherapies, and it could be particularly relevant in the case of patients who experience chronic conditions, such as chronic pain, since they need to turn the trained skills into habits so these can have a meaningful impact on their health. The present systematic review has observed a moderate frequency of home practice during the interventions (around four days per week), although, in terms of time invested, a very high variability has been detected. Despite the high degree of heterogeneity of the studies included in this review, it seems clear that the amount of home practice has a significant impact on different outcomes related to chronic pain since most studies that explored this effect found significant results [41,42,44,47,51,58,62]. This goes in line with a previous systematic review which observed significant effects of home-practice frequency (MBSR or MBCT programs) on different mental health outcomes [28].

When comparing studies conducted on samples with primary and secondary chronic pain, following the IASP categorization [2], it was observed that, on average, the number of days of practice per week was higher in samples with secondary chronic pain (M_Secondary_ = 5.63, SD = 1.11 vs. M_Primary_ = 4.16, SD = 1.25), while patients with primary chronic pain conditions invested more minutes per day (M_Primary_ = 27.37, SD = 13.56 vs. M_Secondary_ = 15.16, SD = 17.69). Although some hypotheses related to the nature of the chronic pain condition and its impact on adherence to meditation exercises could be drawn, it is considered that more studies are needed, not only because of the heterogeneity of samples and interventions but also due to the amount of home practice in primary chronic pain conditions was reported in 18 studies, while only 6 studies were focused on secondary chronic pain, which hinders a precise comparison.

It is worth mentioning that four studies were conducted on samples of adolescents: Ali et al. [62] and Hesse et al. [59] studied a sample of adolescents with primary chronic pain conditions, and Chadi et al. [54,57] tested the effects of an MBI on samples of teenagers with chronic pain as a result of a medical condition. The adherence to home practice in these studies was notable in terms of days per week (ranging from 4 to 6.5 on average), but the amount of time invested each day was generally very low (ranging between 4 and 8 min per day on average). Although the studies reported significant effects of the interventions on treatment outcomes, such as functionality, anxiety, depression, and pain coping, the small sample sizes and some other methodological shortcomings need to be considered when interpreting these results. The review conducted by Lin et al. [64] considered that MBIs could lead to improvement in the overall quality of life for adolescents suffering from chronic pain, but a systematic review [65] concluded that the evidence was inconsistent. Some authors have pointed out that learning mindfulness might require meta-cognitive skills that many adolescents may not have developed yet [66,67], in addition to discipline and consistency regarding meditation practice, which are also uncommon among teenagers. Thus, it is suggested that before implementing “third wave” CBT psychotherapies for adolescents with chronic pain, previous steps addressed at promoting the required basic skills should be taken.

The MBSR and MBCT programs were the most common “third wave” psychotherapies in the reviewed studies (eight and six studies, respectively). In these programs, practicing 6 days a week for approximately 45 min per practice is recommended [16,17]. Four of the reviewed studies using MBCT or MBSR reported both the frequency of practice and the minutes invested per day [26,43,44,58], and none of them reached the recommended adherence; on average, patients practiced 39.6% of the recommended time. This is considerably lower than the result reported in the systematic review conducted by Parsons et al. [28], in which patients with different mental health conditions undergoing MBSR or MBCT practiced around 64% of the recommended time. With due caution, since our finding is based on very few studies—other studies only reported the minutes of practice but not its frequency, it could be hypothesized that patients with chronic pain may present some difficulties in engaging in home meditation practices that should be acknowledged when implementing programs such as MBSR and MBCT: adaptations to posture may be required (e.g., laying down and standing yoga might be too strenuous for patients with pain), as well as shortening the duration of some practices or sessions—as some studies have already tested for other “third wave” psychotherapies [68]—and offering adapted audio guides and other resources to help patients to focus their attention on the exercise when they practice at home. Reducing perceived barriers is key to the individuals engaging in meditation practices in the long term [69], and previous studies conducted on different samples found that “pragmatic barriers” (e.g., being unable to sit for long periods of time, not having enough time or the optimal environment to meditate, difficulty lying on the floor) are common concerns among people with chronic pain who start practicing [70,71], which suggests that the abovementioned adaptations could be useful for other populations besides chronic pain patients.

Among the interventions tested in the reviewed studies, three of them [38,54,63] were eHealth programs, i.e., health services delivered or enhanced through the Internet and related technologies (e.g., telehealth, mobile apps, etc.) [72,73,74]. These studies reported frequent home practice but of short duration: Chadi et al. [54] reported that patients using the eHealth intervention practiced around 6 days per week only for a total of 30 min per week (i.e., an average of 5 min per day), although this was equivalent to the amount of practice reported by the group who received the MBI in face-to-face sessions; Hearn and Finlay [63] observed that most patients (72.2%) adhered to the autonomous practices in their MBI (i.e., 10 min, 6 days a week); and Trompetter et al. [38] reported that the majority of the participants practiced 3 days per week, around 15–20 min per day.

While eHealth may offer patients with chronic pain a number of advantages (e.g., geographic adaptability, schedule flexibility, easy accessibility), some studies using eHealth programs, particularly those without any guidance, highlight the difficulties in achieving adequate adherence rates [75]. Learning how to meditate, as with any other skill, requires at first some guidance on how to perform the practice; moreover, difficulties and doubts are often presented while practicing, for which some external support is considered adequate, at least initially. Some of the reviewed studies have observed a significant association between session attendance, session engagement, and frequency of autonomous practice [52,58], which suggests that a certain degree of guidance may be needed to engage in practices. In addition, a previous systematic review found that, in Internet-delivered psychotherapies, the existence of guidance is associated with better outcomes [76]. Therefore, blended interventions, i.e., a combination of eHealth with face-to-face sessions [77], might have the potential to maintain the advantages of eHealth while offering guidance to patients. A blended MBI was studied by Cebolla et al. [37], who observed that, on average, patients with irritable bowel syndrome practiced around 30 days during the 8-week blended intervention (i.e., around 55% of days), and they invested approximately 15 min per day. Remarkably, half of the surveyed participants reported practicing regularly (between five–seven days per week), which could be considered a notable degree of adherence. In any case, new studies testing the adherence to autonomous practices in blended interventions and their impact on health outcomes are required to discuss their therapeutic potential in chronic pain conditions.

Finally, for what concerns formal vs. informal meditation, two of the three studies which compared them found that informal practices were more common [37,57]. These practices refer to applying mindfulness to day-to-day activities, such as eating or walking, without the need to use guidance or presenting a certain posture, and some studies have observed that this type of practice is associated with improvements in different health outcomes, even more than formal practices [78,79]. These latter practices are considered basic exercises to start developing the “mindful state”, i.e., the ability to observe thoughts and feelings in a non-judgmental, non-attached manner. While some programs, such as MBCT or MBSR, recommend 45 min of formal practices throughout the program, they intertwine these exercises with informal practices from the very first sessions. For patients with chronic pain, who, according to the present systematic review, tend to present with a pattern of moderately frequent low-duration meditations, it could be suitable to shorten the requirements of formal practices while finding ways to potentiate informal practices.

### Limitations and Future Research Directions

The main strengths of this review are that the searches for articles were carried out in different scientific databases and trial registries, verified the selection decisions by three reviewers, and assessed the quality of the study by peers. It is also highlighted that the criteria included RCTs, randomized uncontrolled trials, open trials with pre-post analysis, as well as pilot studies, as long as they followed one of the above study designs, so the risk of not including studies relevant to the objective of the review was. Finally, both primary and secondary chronic pain have been included, following the classification carried out by the IASP [2]. However, there are certain limitations in the present review that should be considered. First, due to the variability of methods in which home practice was recorded and the heterogeneity of how its frequency was reported, it has not been possible to perform a meta-analysis. Moreover, it needs to be noted that the frequency and duration of home practices were mostly self-reported, with the associated inevitable bias. Second, a low risk of bias was concluded in only 10 studies (out of 31), making it necessary to state that higher-quality studies are needed. Third, data on follow-up assessments were only reported in four studies [48,53,56,60], which hindered the extraction of conclusions about the tendency to adhere to home practice once the interventions end. Finally, only studies published in English have been included, which may have implied the loss of articles relevant to the objective of the review.

For future studies, it is considered essential to systematically establish the way in which participants should record what type of practices and the frequency they perform at home (i.e., number of days per week) as well as the duration of each practice. In this regard, clear indications on the recommended amount of home practice should be presented to patients; most studies conducted on this topic do not report presenting such indications [80], and those which do either are based solely on the MBSR and MBCT protocols (6 days per week, 45 min per practice) or establish arbitrary ranges [34,41,42,43,49,53,59,63]. While it seems clear that higher doses of practice are related to better health outcomes, it has not been established which is the adequate nor the minimum amount of home practice associated with improvements. Probably, individual differences play an important role in this regard, but future studies should explore these aspects so that empirically-based recommendations can be offered to patients.

In another vein, new technologies offer the possibility to improve the assessment of home practice: wearable technologies, smartphone apps, text message reminders to fill in practice diaries, or online portals may be used to enhance adherence to home practice and facilitate its evaluation [28,81]; these could be complemented with some biometric variables (e.g., heart rate variability, respiration rate) to achieve a more objective measure of meditation [82], although informal practices will probably remain hard to assess with precision. On the other hand, due to the risk of abandoning the practices once the intervention program has finished, it is essential to include follow-up sessions that favor a higher rate of maintenance and analyze their impact on long-term clinical variables. Another important point is the need to improve the quality and rigor of research designs in future studies; it is indispensable to work with properly calculated sample sizes that favor a better quality of statistical analyses as well as to perform both intention-to-treat and per-protocol analyses. Finally, it should be noted that it is highly relevant to delve into the factors involved in adherence to home practice in “third wave” CBT psychotherapies and, thus, increase the rates of practice time in general [83].

## 5. Conclusions

The present systematic review has observed that patients with chronic pain present a pattern of moderately frequent—around four days per week—the autonomous practice of mediation while undergoing “third wave” psychotherapies, with a high degree of variability in terms of duration (i.e., minutes per practice). The reviewed studies support the relationship between the amount of practice and the treatment outcomes. The MBSR and MBCT programs were the most common interventions, and although data on practice at home was incomplete in many studies, it seems that patients with chronic pain practice significantly less than what is recommended in these programs (i.e., 6 days a week, 45 min per day), maybe due to some difficulties (e.g., maintaining a posture or focusing attention for long periods of time) that could justify some adaptations in the way that home practices are prescribed and supported. In a similar line, adolescents with chronic pain, who usually engage in very short practices, may require some previous development of meta-cognitive skills so that they can adhere properly to MBIs. In this review, a few eHealth interventions were included, with heterogeneous adherence rates to home practice; it is considered that alternatives that include guidance (e.g., blended interventions) could be more suitable for chronic pain patients, also considering some adaptations, such as shortening or reducing formal practices during the intervention in favor of informal practices which seem to be more easily implemented and even more impactful.

## Figures and Tables

**Figure 1 ijerph-20-04438-f001:**
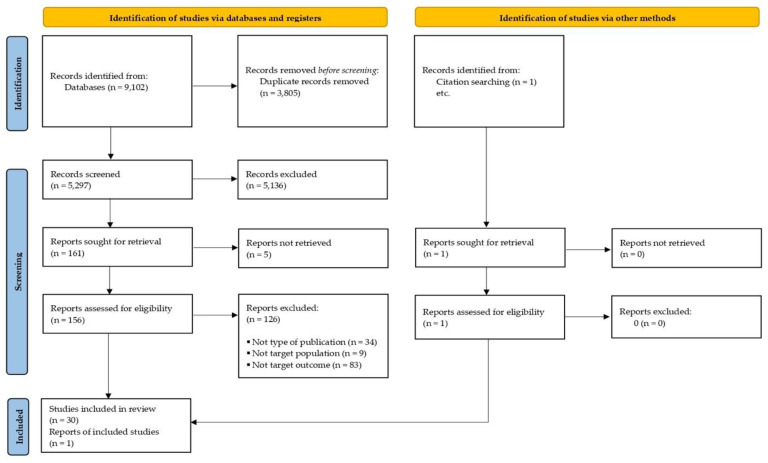
PRISMA diagram of study flow.

**Table 2 ijerph-20-04438-t002:** Frequency of home practice during the intervention.

Authors (Year)	Days of Practice	Minutes of Practice	Observations
Ali et al. (2017) [62]	-	Median of 434 min (range: 0–1736) in 8 weeks	Only one subject reported not having practiced at home. The amount of home practices was associated with improvement: subjects with greater amounts of home practice over 8 weeks (top 50% of sample) had 44% improvements in functionality and 26% improvements in impact scores. Subjects with lower amounts of home practice (bottom 50% of sample) had only 4% improvements in functionality scores and 9% worsening in impact scores.
Carson et al. (2010) [42]	-	Total practice (daily): M = 40 (range 11–97) Postures (daily): M = 19 (range 4–57) Meditation (daily): M = 13 (range 2–29) Breathing exercises (daily): M = 8 (range 2–16)	More practice was associated with greater overall improvement in symptoms and more improvement in fatigue and relaxation.
Cebolla et al. (2021) [37]	M = 31 days (SD = 12.04) during the 8 weeks	M (daily) = 16 (SD = 9.49)	A subsample of 19 patients, who were interviewed after completing the intervention, reported spending more time on informal practices than on formal meditation. In addition, half of them practiced regularly (between 5–7 days per week).
Chadi et al. (2016) [57]	M = 4 (range 1–10) per week	M (daily) = 8 (range 1–>25)	The most common pattern of practice was 4 to 6 times per week, between 6–15 min per practice. Informal practices were the most frequent (e.g., mindfulness while brushing teeth, waiting for the bus), followed by formal sitting/walking meditations, body scans, and mindful eating/breathing.
Chadi et al. (2018) [54]	M_In-person_ = 6.5 (range 1.4–13.4) per week M_Online_ = 6 (range 2.9–9.7) per week	M_In-person_ (weekly) = 28.8 (range 4.3–154.7) M_Online_ (weekly) = 30.6 (range 6.6–107.8)	Levels of self-reported home practices were equivalent in both groups. Reported practices included all nine types of practices taught during MARS-A sessions (i.e., breathing meditation, body scan, mindful movement, eating meditation).
Cooperman et al. (2021) [49]	M = 31 days (55.6% of days, SD n.r.) during the 8 weeks	M (daily) = 23.8 (SD = 25.3)	-
Day et al. (2014) [58]	M_Completers_ = 5.06 (SD = 1.19) per week M_Dropouts_ = 1.58 (SD = 1.39) per week	M_Completers_ (weekly) = 190.8 (SD = 47.4) M_Dropouts_ (weekly) = 37.2 (SD = 42)	While controlling for attendance, completers (i.e., attended at least 4 sessions) practiced significantly more days on average and more time (in minutes) than dropouts. Compared to TAU, the MBCT completers significantly improved in pain catastrophizing, pain acceptance, and headache management self-efficacy (while these changes were not observed when comparing the whole sample).
Day et al. (2016) [52]	-	M (8 weeks) = 1301 (SD = 666)	Higher ratings of in-session engagement (therapist rated) were associated with higher attendance and a greater amount of time spent on at-home meditation practice.
Day et al. (2020) [55]	-	M_MM_ (8 weeks) = 3025 (SD = 242) M_MBCT_ (8 weeks) = 2088 (SD = 219)	Both MM and MBCT groups presented significantly higher amounts of at-home practice compared to the CT group (M = 715 min; SD = 224 min). The amount of practices in MBCT was significantly lower than in MM (P = 0.006). For both mindfulness conditions (MM and MBCT), engagement in the 3-min breathing space accounted for more time in meditation than engagement in the extended, formal meditation. The models testing the Amount of At-home Practice × Treatment Condition interaction found nonsignificant, medium-effect-sized interactions for both pain interference and pain intensity.
Donnino et al. (2021) [48]	-	M_4 weeks_ (weekly) = 580 (SD = 680.8) M_8 weeks_ (weekly) = 585 (SD = 708.3) M_13 weeks_ (weekly) = 315 (SD = 490.3) M_26 weeks_ (weekly) = 335 (SD = 398.2)	Most patients spent between 1 and 6 h per week practicing skills taught during the intervention (73%), with 2 to 4 h being the most common response in MBSR (chosen 42% of the time). The MBSR group presented higher frequency of practice compared to the PSRT group.
Gardiner et al. (2020) [36]	M = 19 days (range 1–63) during the 9 weeks	M_Body scan_ (9 weeks) = 61 (SD = n.r.) M_Other meditations_ (9 weeks) = 25 (SD = n.r.)	-
Garland et al. (2014) [45]	-	M (weekly) = 166.9 (SD = 93.4)	There were no significant between-group differences in duration of weekly homework practice.
Greenberg et al. (2019) [50]	-	-	3RP + Get active: 4 out of 6 participants (66%) completed ≥ 5 out of 7 weeks of homework (acceptable) 3RP + Get active with Fitbit: 6 out of 7 participants (86%) completed ≥ 5 out of 7 weeks of homework (excellent)
Hearn and Finlay (2018) [63]	M = 6 days per week	M_Completers_ (8 weeks) = 960 (SD = 0)	The course delivered two 10-min audio-guided meditations each day, on 6 out of 7 days a week, for 8 weeks. Those, who dropped out of mindfulness training (n = 10), completed an average of 217 min of practice (range 40–460 min).
Hesse et al. (2015) [59]	M = 4.69 (SD = 1.84) per week		Participants in this study were directed to listen to 10–15-min guided mindfulness recordings at least once each day or to practice without guidance.
Howarth et al. (2019) [60]	M_Week 1_ (weekly) = 4.58 (SD = 1.61) M_Month 1_ (monthly) = 8.50 (SD = 4.98)	-	Patients were free to use the “body scan” audio-guide (10 min) as many times as they wanted for 4 weeks.
Johannsen et al. (2018) [51]	-	-	More homework practice during the 8-week program predicted increases in mindfulness non-reactivity (*p* = 0.02, d = 0.70), but did not predict reductions in pain catastrophizing over time. Total minutes of home practice during the previous week at T2–T4 did not predict changes in mindfulness non-reactivity or changes in pain catastrophizing over time.
Mittal et al. (2022) [35]	-	M (8 weeks) = 1468.8 (range 1144.8–1944)	Participants were invited to practice at home for 45 min for 6 days each week (2160 min in total). Participants reported frequency of formal meditation practice being 68% (range 53–90%).
Morone et al. (2008) [43]	M = 4.3 (range 0–7) per week	M (daily) = 31.6 (range 0–52)	The recommendations were daily meditations (six of seven days/week) lasting 50 min (45 min of meditation, 5 min to complete a diary). Nineteen participants reported that they continued to meditate in the 3-month follow-up assessment.
Pérez-Aranda et al. (2019) [26]	M = 2.5 (SD = 2.17) per week	M (daily) = 53 (SD = 49)	Twenty-three participants (30.7%) reported no practice any day during the intervention, and 41 (54.7%) reported practicing at least 2 days per week.
Pradhan et al. (2007) [61]	-	M (8 weeks) = 2827 (SD = 1276)	Participants reported practicing over an hour a day for 6 days a week during the program. Of 49 possible days during which practice could have been undertaken, the median was 42 (interquartile range: 40–48). At the 6-month follow-up assessment, 85.7% of participants evaluated reported undertaking MBSR practices in the 2 weeks before the visit. Neither overall sum of practice time nor sum of time spent on a specific practice predicted change in any measure by 2 months. However, from baseline to 2 months, each 1-day increase in practice was associated with improvement of –0.03 in depressive symptoms and −0.01 in psychological distress.
Rae et al. (2020) [46]	M = 2.52 (SD = n.r.) per week	M (per practice) = 20.37 (SD = n.r.)	The amount of practice (both in terms of days per week and minutes per practice) was very similar in the control group. There were 4 participants in the yoga group who averaged 3 or more days of home practice and 30 min or more of practice per session, 6 people who averaged over 20 min, and 2 who did no home practice at all.
Rosenzweig et al. (2010) [44]	M = 6 (SD = 3) per week	M (per practice) = 20 (SD = 14)	Among the 41 participants who had self-reported data on frequency of practice, greater average weekly home meditation practice was significantly associated with greater reduction in overall psychological distress and somatization symptoms, as well as with an increase in self-rated general health.
Seng et al. (2019) [56]	M_Treatment_ (8 weeks) = 31.20 (SD = 14.30) M_Post-treatment_ (4 weeks) = 14.90 (SD = 9.70)	-	Participants in the MBCT group practiced mindfulness on 980/1327 (73.9%) of recorded diary days during the treatment period; 2 participants never recorded a mindfulness practice.
Trompetter et al. (2014) [38]	-	-	Mindfulness exercises were most often performed for 3 days per week (by 21% of participants) for 15–20 min (by 36% of participants). In total, 48% (n = 39) of participants adhered to ACT, with adherence defined as completing the intervention and working with the ACT intervention for ≥3h per week. After 6 months, 77% of ACT participants reported to have incorporated mindfulness exercises into their daily life.
Van Gordon et al. (2017) [47]	-	M (daily) = 41.11 (SD = 15.26)	Results showed significant linear relationships between the number of minutes meditated and all outcome differences (i.e., functionality, pain, psychological distress, sleep quality, and non-attachment).
Wong et al. (2011) [39]	M = 3.6 (SD=2.02) per week	-	The difference in practicing time of the 2 groups was not significant. In the follow-up assessment (6 months after the intervention finished), 65% of the participants in the MBSR group claimed that they were still practicing meditation.
Zanca et al. (2022) [34]	-	M (weekly) = 98 (SD = n.r.)	Participants in the control group spent approximately half as much time as CMI group participants in home practice (52 min/week).
Zernicke et al. (2012) [40]	-	M (weekly) = 137 (SD = n.r.)	Attempts to collect data on adherence to meditation practice over the follow-up period were not successful and it was not possible to assess any associations between further changes and home practice.
Zgierska et al. (2016) [41]	M_Formal Practice, Week 1–8_ = 5.1 (SD = 2.1) M_Formal Practice, Week 9–26_ = 4.1 (SD = 2.6) M_Informal practice, Week 1–8_ = 4.9 (SD = 2) M_Informal Practice, Week 9–26_= 4.2 (SD = 2.5) per week	M_Formal Practice, Week 1–8_ = 188.3 (SD = 94.4) M_Formal Practice, Week 9–26_ = 153.3 (SD = 139.6) M_Informal practice, Week 1–8_ = 110.1 (SD = 78.8) M_Informal Practice, Week 9–26_= 99.5 (SD = 131.3)	Those reporting on average at least 150 min of formal practice per week (>80% of the study-recommended 180 min/week ‘‘dose’’) during at least two thirds of the study periods were defined as ‘‘consistent’’ meditators (n = 10), while the remaining participants (n = 11) were classified as ‘‘inconsistent’’ meditators. ‘‘Consistent’’ meditators maintained a stable level of formal and informal practices, whereas ‘‘inconsistent’’ meditators showed a significant decline in both formal and informal practices over time; these subgroups did not differ for baseline characteristics, session attendance, or treatment satisfaction ratings.
Zgierska et al. (2016) [53]	-	M_Total sample_ (weekly) = 164 (SD = 122.1) M_Consistent_ (weekly) = 256.2 (SD = 102.7) M_Inconsistent_ (weekly) = 71.6 (SD = 44.8)	Participants were categorized as either “consistent” meditators (≥150 min/week of formal meditation practice during at least 2/3 of the study) or “inconsistent” meditators (<150 min/week of practice during at least 2/3 of the study); both subgroups did not differ in their attendance in the intervention sessions. There were no statistically significant differences in the change in self-reported and biomarker measures between the consistent and inconsistent meditators; however, when compared with controls, the consistent meditators had a greater decrease in pain ratings.

n.r. = “not reported”.

## Data Availability

Not applicable.

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
