# Peer review of "A Systematic Review of the Adherence to Home-Practice Meditation Exercises in Patients with Chronic Pain"

_ijerph, 2023, doi:10.3390/ijerph20054438_

Round 1
Reviewer 1 Report
This systematic review is focused on adherence to meditation among chronic pain patients. This is a well-designed and well-written manuscript and can be accepted in “as is” form. Below are two minor suggestions:
Please edit the first sentence of the abstract: “Third wave” psychotherapies are effective for treating chronic pain conditions.” since many readers might not be familiar with the term “third wave”.
Consider adding this reference MacPherson et al Can J Pain 2022 (https://www.ncbi.nlm.nih.gov/pmc/articles/PMC9176230/) when describing eHealth technologies for chronic pain that deliver mindfulness and CBT interventions.
Author Response
This systematic review is focused on adherence to meditation among chronic pain patients. This is a well-designed and well-written manuscript and can be accepted in “as is” form. Below are two minor suggestions:
Please edit the first sentence of the abstract: “Third wave” psychotherapies are effective for treating chronic pain conditions.” since many readers might not be familiar with the term “third wave”.
Response:Thank you so much for your comments and suggestions.
We have edited the first sentence of the abstract so that the psychotherapies to which we refer in the article are specified, and misunderstandings are avoided (line 22):
Mindfulness-, compassion- and acceptance-based (i.e., “third wave”) psychotherapies…
Consider adding this reference MacPherson et al Can J Pain 2022 (https://www.ncbi.nlm.nih.gov/pmc/articles/PMC9176230/) when describing eHealth technologies for chronic pain that deliver mindfulness and CBT interventions.
Response:Thank you for suggesting this relevant work. We have added it in line 419.
Reviewer 2 Report
Thank editor to give me an opportunity to review this work. Authors propose a systematic review of the adherence to home-practice medita-tion exercises in patients with chronic pain.
The objective of this systematic review aimed at evaluating the frequency, duration, and effects of home practice in pa-tients with chronic pain undergoing a “third wave” psychotherapy.
I believe this paper will make an important contribution to the literature.
The study has many strengths, including addressing a question of interest to the readership of this journal.
A major limitation of the study is due to the variability of methods in which home practice was recorded and the heterogeneity of how its frequency was reported, therefore the authors did not perform a meta-analysis.
I have a number of thoughts on how the presentation of the study might be revised, as discussed below.
Regarding the major comments:
1. Authors should explain more details about the important variables.
The literature review forma very confuse for reader which need more clarify demonstration it.
I ask to consult and add the following studies:
- https://doi.org/10.1515/sjpain-2022-0107
- https://doi.org/10.2147/PRBM.S44762
- https://doi.org/10.3390/ijerph18137176
- https://doi.org/10.1016/j.jpain.2019.04.009
2. The results are well presented through analysis. However, discussion and argumentation need to explain more details about each important factor. Therefore, this part should be improved because this study found several interesting points but less explanation about them.
3. Make the discussion sound to support the results according to the study's objectives.
4- Writing needs improvements as grammatical mistakes and redundant expressions are observed throughout the manuscript.
minor revision:
Figure 1 is not sharp. The figure needs high-definition resolution.
Author Response
Thank editor to give me an opportunity to review this work. Authors propose a systematic review of the adherence to home-practice medita-tion exercises in patients with chronic pain.
The objective of this systematic review aimed at evaluating the frequency, duration, and effects of home practice in pa-tients with chronic pain undergoing a “third wave” psychotherapy.
I believe this paper will make an important contribution to the literature.
The study has many strengths, including addressing a question of interest to the readership of this journal.
A major limitation of the study is due to the variability of methods in which home practice was recorded and the heterogeneity of how its frequency was reported, therefore the authors did not perform a meta-analysis.
I have a number of thoughts on how the presentation of the study might be revised, as discussed below.
Regarding the major comments:
- Authors should explain more details about the important variables.
The literature review forma very confuse for reader which need more clarify demonstration it.
I ask to consult and add the following studies:
- https://doi.org/10.1515/sjpain-2022-0107
- https://doi.org/10.2147/PRBM.S44762
- https://doi.org/10.3390/ijerph18137176
- https://doi.org/10.1016/j.jpain.2019.04.009
Response:Thank you for your comments and suggestions to improve the quality of our work.
Following the reviewer’s suggestion, we have carefully reviewed the Introduction, trying to simplify some parts and including more information on important variables such as psychological flexibility using, among others, the recommended articles, which are undoubtedly relevant to the topic that is being discussed (lines 71, 83):
These interventions are not primarily focused on symptom relief, but instead, on the promotion of wellbeing. This is done through the cultivation of psychological flexibility (i.e., a wide range of abilities such as recognizing and adapting to situational demands; shifting mindsets or behavioral repertoires when these strategies compromise personal or social functioning; maintaining balance between life domains; and being aware, open, and committed to behaviors that are congruent with deeply held values) [14,15]. Meditation practices play a central role in these programs since they train the patient to interact with their thoughts and feelings in a more detached manner.
Such interventions have been categorized under the term of “third wave” CBT psychotherapies: Mindfulness-Based Stress Reduction (MBSR) [16], Mindfulness-Based Cognitive Therapy (MBCT) [17] and Acceptance and Commitment Therapy (ACT) [18] are some examples. Previous studies have concluded that these interventions’ effects on reducing pain interference are achieved through enhancing pain acceptance and psychological flexibility [19], as well as reducing fear of pain, pain-related anxiety, and fear-avoidance beliefs [20]. These changes play a significant role in improving functional impairment, anxiety, depression, and psycho-logical distress in chronic pain conditions [21–25].Some studies report higher effects of “third wave” psychotherapies than multicomponent interventions [26], although the superiority of these interventions compared to classic CBT is still unclear [27].
- The results are well presented through analysis. However, discussion and argumentation need to explain more details about each important factor. Therefore, this part should be improved because this study found several interesting points but less explanation about them.
Response:We believe the Results section presents the main variables that conform the outcomes of this systematic review: frequency of practice (i.e., number of days), duration of the practices (i.e., minutes), and (when reported) associations between amount of practice and intervention’s effects. In our opinion, although some other factors could be considered important (e.g., barriers and facilitators), since they were generally not reported in the reviewed studies, we have not granted them major attention in the Results section (although they are now commented with more detail in the discussion).
- Make the discussion sound to support the results according to the study's objectives.
Response:The discussion has been carefully reviewed by the authors and now includes new information on different topics that, in our opinion, strengthen our argument and are related to the objectives of this work; first, the role of barriers to meditation practice in lines 410:
Reducing perceived barriers is key so that individuals engage with meditation practices in the long term [69], and previous studies conducted on different samples found that “pragmatic barriers” (e.g., being unable to sit for long periods of time, not having enough time or the optimal environment to meditate, difficulty lying on the floor) are common concerns among people with chronic pain who start practicing [70,71], which suggests that the abovementioned adaptations could be useful for other populations besides chronic pain patients.
Also, we have delved into the importance of reporting the recommended amount of practice on each study, and how future studies should explore empirically how much that should be rather than using previously determined indications (MBSR or MBCT’s 45 minutes per 6 days a week) or applying a random criterion (lines 488):
In this regard, clear indications on the recommended amount of home practice should be presented to patients; most studies conducted on this topic do not report presenting such indications [80], and those which do either are based solely on the MBSR and MBCT protocols (6 days per week, 45 minutes per practice) or establish arbitrary ranges [34,41–43,49,53,59,63]. While it seems clear that higher doses of practice are related to better health outcomes, it has not been stablished which is the adequate nor the mini-mum amount of home practice that is associated to improvements. Probably, individual differences play an important role in this regard, but future studies should explore these aspects so that empirically-based recommendations can be offered to patients.
Finally, we have also referred to what design aspects should be considered with caution in future studies to minimize risk of bias (lines 507):
Another important point is the need to improve the quality and rigor of research designs in future studies; it is indispensable to work with properly calculated sample sizes that favor a better quality of statistical analyses as well as to perform both intention-to-treat and per-protocol analyses.
4- Writing needs improvements as grammatical mistakes and redundant expressions are observed throughout the manuscript.
Response:The article has been edited by an English-native coauthor.
minor revision:
Figure 1 is not sharp. The figure needs high-definition resolution.
Response:We have improved the quality of the image, although we rely on the journal’s assistant editors to perform further modifications to it in case it is needed once the article is published.
Reviewer 3 Report
Thanks for the opportunity to review this systematic review by Barceló-Soler et al. Overall, I have no significant concerns about the review. It is well written and presents an important issue that needs to be addressed in these studies - the frequency (i.e., number of days) and amount of practice (i.e., minutes per day) of at-home practice of various meditation interventions.
Given the heterogeneity of the studies especially related to outcome reporting, a meta-analysis was not possible. This review does provide a nice overview and presents the current gaps in studies using meditation. Well done.
I have some general recommendations for the review.
(1) Risk of Bias (Tables S1 and S2). Please provide a listing of the different risk categories. Also, at the end of the manuscript, it would be helpful to list study design considerations to improve the rigor of these studies. How can we move this field further?
(2) Barriers to at-home adherence. It would be helpful to discuss if barriers to or facilitators adherence. It was mentioned as a secondary outcome. If these are not adequately discussed in the cited papers, it might be possible to extrapolate from meditation studies for mental health.
(3) Gold standard for meditation. In the review, it appears that studies use (if at all) different criteria for recommended adherence either by frequency and duration. Is there empirical support (or did they provide support) for this criteria? This might be an important topic to address. What about the meditation studies for mental health?
(3) Would it be possible to identify (by subgroup) why some of the papers were excluded (Figure 1) from different stages in the analysis?
Author Response
Thanks for the opportunity to review this systematic review by Barceló-Soler et al. Overall, I have no significant concerns about the review. It is well written and presents an important issue that needs to be addressed in these studies - the frequency (i.e., number of days) and amount of practice (i.e., minutes per day) of at-home practice of various meditation interventions.
Given the heterogeneity of the studies especially related to outcome reporting, a meta-analysis was not possible. This review does provide a nice overview and presents the current gaps in studies using meditation. Well done.
I have some general recommendations for the review.
(1) Risk of Bias (Tables S1 and S2). Please provide a listing of the different risk categories. Also, at the end of the manuscript, it would be helpful to list study design considerations to improve the rigor of these studies. How can we move this field further?
Response:Thank you very much for the review and comments made to improve the quality of our manuscript.
We agree with the reviewer that the risk categories needed to be presented along with the risk of bias tables. These are now included in Table S1 and S2.
Also, following the reviewer’s recommendation, we have acknowledged in the Limitations section some aspects related to study design that should be considered when conducting future studies (lines 507):
Another important point is the need to improve the quality and rigor of research designs in future studies; it is indispensable to work with properly calculated sample sizes that favor a better quality of statistical analyses as well as to perform both intention-to-treat and per-protocol analyses.
(2) Barriers to at-home adherence. It would be helpful to discuss if barriers to or facilitators adherence. It was mentioned as a secondary outcome. If these are not adequately discussed in the cited papers, it might be possible to extrapolate from meditation studies for mental health.
Response: Following the reviewer’s suggestion, we have opted by including information on barriers found in studies conducted on different samples since the studies included in the systematic review did not provide enough data on this (lines 410):
Reducing perceived barriers is key so that individuals engage with meditation practices in the long term [69], and previous studies conducted on different samples found that “pragmatic barriers” (e.g., being unable to sit for long periods of time, not having enough time or the optimal environment to meditate) are common concerns among people who start practicing [70,71], which suggests that the abovementioned adaptations could be useful for other populations besides chronic pain patients.
(3) Gold standard for meditation. In the review, it appears that studies use (if at all) different criteria for recommended adherence either by frequency and duration. Is there empirical support (or did they provide support) for this criteria? This might be an important topic to address. What about the meditation studies for mental health?
Response: As the reviewer points out, recommendations on the amount of practice (both in terms of days per week and in minutes per practice) are notably variable and, often, not empirically-based. We had noted it in the Limitations section (lines 486):
For future studies, it is considered essential to systematically establish the way in which participants should record what type of practices and the frequency they perform at home (i.e., number of days per week), as well as the duration of the practice and the recommendations that patients receive in this regard.
Nevertheless, we agree that this is an important topic and deserves more attention, so this section has been extended (lines 488):
In this regard, clear indications on the recommended amount of home practice should be presented to patients; most studies conducted on this topic do not report presenting such indications [80], and those which do either are based solely on the MBSR and MBCT protocols (6 days per week, 45 minutes per practice) or establish arbitrary ranges [34,41–43,49,53,59,63]. While it seems clear that higher doses of practice are related to better health outcomes, it has not been stablished which is the adequate nor the minimum amount of home practice that is associated to improvements. Probably, individual differences play an important role in this regard, but future studies should explore these aspects so that empirically-based recommendations can be offered to patients.
(3) Would it be possible to identify (by subgroup) why some of the papers were excluded (Figure 1) from different stages in the analysis?
Response:Following the reviewer’s suggestion, we have included in the flowchart the count of papers excluded by subgroups in the section “Reports excluded” by eligibility criteria (Figure 1).